# Promising Therapeutic Efficacy of GC1118, an Anti-EGFR Antibody, against KRAS Mutation-Driven Colorectal Cancer Patient-Derived Xenografts

**DOI:** 10.3390/ijms20235894

**Published:** 2019-11-24

**Authors:** Hye Won Lee, Eunju Son, Kyoungmin Lee, Yeri Lee, Yejin Kim, Jae-Chul Lee, Yangmi Lim, Minkyu Hur, Donggeon Kim, Do-Hyun Nam

**Affiliations:** 1Department of Anatomy and Cell Biology, Sungkyunkwan University School of Medicine, Suwon 16149, Korea; nsproper@skku.edu; 2Single Cell Network Research Center, Sungkyunkwan University, Suwon 16149, Korea; 3Institute for Refractory Cancer Research, Samsung Medical Center, Seoul 06351, Korea; ejs@1stbio.com (E.S.); kyoungmin@g.skku.edu (K.L.); yeri.lee26@gmail.com (Y.L.); yiejin89@g.skku.edu (Y.K.); 4Department of Health Science and Technology, Samsung Advanced Institute for Health Science and Technology, Sungkyunkwan University, Seoul 06351, Korea; 5Research Institute for Future Medicine, Samsung Medical Center, Seoul 06351, Korea; 6Translational Research 1 Team, MOGAM Institute for Biomedical Research, Yongin 16924, Korea; jclee@mogam.re.kr (J.-C.L.); ymlim@mogam.re.kr (Y.L.); zymogen@mogam.re.kr (M.H.); 7Department of Neurosurgery, Sungkyunkwan University School of Medicine, Samsung Medical Center, Seoul 06531, Korea

**Keywords:** colorectal cancer, patient-derived xenograft, EGFR-targeting therapeutic antibody, *KRAS* mutation, PI3K/mTOR/AKT inhibitor

## Abstract

Epidermal growth factor receptor (EGFR)-targeted monoclonal antibodies, including cetuximab and panitumumab, are used to treat metastatic colorectal cancer (mCRC). However, this treatment is only effective for a small subset of mCRC patients positive for the wild-type *KRAS* GTPase. GC1118 is a novel, fully humanized anti-EGFR IgG1 antibody that displays potent inhibitory effects on high-affinity EGFR ligand-induced signaling and enhanced antibody-mediated cytotoxicity. In this study, using 51 CRC patient-derived xenografts (PDXs), we showed that *KRAS* mutants expressed remarkably elevated autocrine levels of high-affinity EGFR ligands compared with wild-type *KRAS*. In three *KRAS*-mutant CRCPDXs, GC1118 was more effective than cetuximab, whereas the two agents demonstrated comparable efficacy against three wild-type *KRAS* PDXs. Persistent phosphatidylinositol-3-kinase (PI3K)/AKT signaling was thought to underlie resistance to GC1118. In support of these findings, a preliminary improved anti-cancer response was observed in a CRC PDX harboring mutated *KRAS* with intrinsically high AKT activity using GC1118 combined with the dual PI3K/mammalian target of rapamycin (mTOR)/AKT inhibitor BEZ-235, without observed toxicity. Taken together, the superior antitumor efficacy of GC1118 alone or in combination with PI3K/mTOR/AKT inhibitors shows great therapeutic potential for the treatment of *KRAS*-mutant mCRC with elevated ratios of high- to low-affinity EGFR ligands and PI3K-AKT pathway activation.

## 1. Introduction

At initial diagnosis, approximately 20% of colorectal cancer (CRC) patients present with distant dissemination, which is associated with a high mortality rate, highlighting the importance of effective systemic therapeutic strategies [1,2]. Commonly-affected signaling pathways include the Wnt and receptor tyrosine kinase (RTK) pathways, the components of which include epidermal growth factor receptor (EGFR), vascular endothelial growth factor, and insulin-like growth factor 1 receptor (IGF1R) [3]. Currently, only 10 drugs, either administered as a monotherapy or in combination, have been approved for use against metastatic CRC (mCRC) [4]. Although integrated multi-omics approaches have improved our understanding of the underlying molecular pathophysiology of mCRC, there is a need to customize treatment strategies to account for the high inter/intra-tumor heterogeneity and the involvement of diverse drivers of mCRC [3,5].

EGFR-family hetero-dimerization, ligand affinity, and signaling cross-talk influence cellular outcomes [6,7]. For example, different binding affinities of various ligands for EGFR result in different levels of tumor growth in CRC cell lines [8]. Such ligands are classified as high- or low-affinity EGFR ligands. High-affinity ligands include epidermal growth factor (EGF), transforming growth factor α (TGF-α), heparin-binding EGF-like growth factor (HB-EGF), and betacellulin (BTC). Low-affinity ligands include amphiregulin (AREG) and epiregulin (EREG) [6]. The unique effects of anti-EGFR monoclonal antibodies (MoAbs), including cetuximab and panitumumab, on mCRC treatment are increasingly being recognized. MoAbs compete with ligands to block downstream signaling by promoting receptor internalization, antibody-dependent cellular cytotoxicity (ADCC), and endocytosis-mediated cytotoxicity; however, acquired resistance to such MoAbs occurs over time [4,9]. The EGFR signaling cascade leads to the activation of various transcription factors that modulate proliferation, migration, angiogenesis, and metastatic spread in mCRC, via three major pathways, namely rat sarcoma (RAS)–rapidly accelerated fibrosarcoma (RAF)–mitogen-activated protein kinase (MAPK), phosphatidylinositol 3-kinase (PI3K)–AKT–mammalian target of rapamycin (mTOR), and Janus kinase/signal transducers and activators of transcription [10,11]. Notably, these pathways have also been implicated in mechanisms of resistance to antibody-mediated EGFR inhibition [10,11,12]. Interestingly, activating mutations in the KRAS proto-oncogene GTPase (*KRAS*) are most common among CRCs, comprising approximately 35%–45% of alterations (point mutations in exons 2, 3, and 4) [12,13,14,15], and these predict primary resistance to anti-EGFR MoAbs, such as cetuximab and panitumumab [16,17,18,19]. This is because constitutively activated RAS downstream signaling can activate multiple processes involved in tumor progression without the influence of EGFR and related receptor kinases [5,10]. There is also circumstantial evidence to suggest that an excess of high-affinity ligands drives resistance to cetuximab [6,8,20,21].

GC1118 is a human anti-EGFR IgG1 antibody that differs from existing anti-EGFR MoAbs, such as cetuximab and panitumumab, in its constant region, affinity, mode of action, and efficacy [8,20]. A recent first-in-human phase I study of GC1118 conducted on patients with refractory solid tumors, including gastric cancer and CRC, showed promising clinical antitumor efficacy and tolerability [22]. Notably, GC1118 exhibited superior inhibitory effects on high-affinity ligand-induced signaling in CRC and gastric cancer cells, regardless of KRAS status, triggering more potent antitumor activity than cetuximab and panitumumab [8,20]. However, persistent activation of the phosphatidylinositol-4,5-bisphosphate 3-kinase catalytic subunit alpha (PIK3CA)/phosphatase and tensin homolog (PTEN) pathway, one of the major downstream pathways, might lead to the activation of EGFR-independent downstream signaling pathways [10], suggesting that the combination of PI3K/mTOR/AKT inhibitors with EGFR MoAbs might be efficacious [23].

The translationally-relevant CRC patient-derived xenograft (PDX) platform, which maintains a high degree of genetic and transcriptional fidelity compared to respective parental tumors, coupled with bioinformatics and high-throughput drug screening, is effective to investigate heterogeneous mCRC with the aim of uncovering novel therapeutic agents [24,25,26]. Herein, we investigated the autocrine expression levels of high- and low-affinity EGFR ligands using a large panel of CRC patient-derived xenografts (PDXs). We also evaluated the therapeutic efficacy of GC1118 alone or in combination with the dual PI3K/mTOR inhibitor BEZ-235 [27], while considering the presence of KRAS mutations and the expression pattern of EGFR ligands (Figure 1A). Figure 1B and Appendix A summarize the clinical and genomic baseline characteristics of 51 stratified CRC patients used to establish a PDX series.

## 2. Results

### 2.1. Genomic Characterization of CRC PDX Models and Expression Levels of High- and Low-Affinity EGFR Ligands According to KRAS Status

All patients underwent excisional biopsy of a primary CRC (*n* = 30, 58.8%) or metastatic lesions (*n* = 21, 41.2%) **(**Figure 1B, left panel). Fourteen (27.5%) and 37 CRC patients (72.5%) were diagnosed with localized (stage I–III) and metastatic disease (stage IV), respectively (Figure 1B, left panel). The primary tumor was in the right colon (cecum to proximal transverse) in 11 cases (21.6%) and the left colon (distal transverse to rectum) in 39 (76.5%) cases. In one case, the location was unknown (*n* = 1 and 2%) (Figure 1B, upper right panel). In general, *KRAS* gene mutations are predominant among *RAS* family gene alterations in mCRC (85%), and approximately 90% of *KRAS* mutations occur within codons 12 and 13 [28]. Here, *KRAS* mutations were observed in 24 (42.1%) cases (Figure 1B, lower right panel), whereas no gene alterations were present in B-Raf proto-oncogene serine/threonine kinase (*BRAF*) (Figure 1B, left panel). PIK3CA and tumor protein P53 (*TP53*) mutations were also detected in seven (16.4 %) and 23 (45.1%) patients (Appendix A), respectively.

Low-affinity ligands EREG and AREG are predominant in CRC, whereas only a small fraction of high-affinity ligands is expressed [29]. Low expression levels of AREG and EREG associated with *KRAS* mutations might indicate a tumor that is less dependent on EGFR and is therefore particularly prone to developing resistance to anti-EGFR MoAbs [6,8,10,20,21]. Moreover, the expression levels of AREG and EREG were found to be significantly decreased in mutant-*KRAS* cases, compared to those in the wild-type cases [30]. Sustained extracellular signal–regulated kinases (ERK) signaling mediated by *KRAS* mutations was shown to boost secretion of the high-affinity EGFR ligands HB-EGF and TGF-α, which in turn activated EGFR in an autocrine fashion [31]. The total expression level of each EGFR ligand (nM) did not show any significant association with *KRAS* mutations as evaluated by ELISA (Appendix A). Notably, consistent with previous reports [30,31], we found that *KRAS*-mutant PDXs tended to show significantly higher fractions of high-affinity EGFR ligands and lower fractions of low-affinity EGFR ligands (Figure 2A,B), in addition to a higher ratio of high- to low-affinity EGFR ligands, than did the *KRAS* wild-type PDXs (Figure 2C,D). This indicates that the distribution of high- and low-affinity EGFR ligands depends on the presence of a *KRAS* mutation.

### 2.2. GC1118 is More Active Than Cetuximab against KRAS-Mutant CRC PDXs

To compare the effectiveness of GC1118 and cetuximab in vivo, 6 CRC PDXs (three *KRAS* wild-types and three *KRAS* mutants; all *PIK3CA* wild-type) were treated with GC1118 for at least 28 days (Figure 3).

To evaluate the effects of GC1118 and cetuximab, tumor growth inhibition index (TGII) values were calculated from the average volume of the treated (Vt) and vehicle control (Vvc) groups using the following equation: TGII (%) = (Vt final -Vt initial)/(Vvc final -Vvc initial) × 100 [32]. For example, if the treatments resulted in no change in growth vs. vehicle-treated controls, TGII (%) = 100. If GC1118 or cetuximab resulted in 70% tumor growth compared to vehicle-treated control tumors, TGII (%) = 300. Both GC1118 (TGII = −36.8%, *p* = 0.0053) and cetuximab (TGII = −29.4%, *p* = 0.006) induced complete tumor regression in CRC-003T PDXs (*KRAS*-wild-type; high-affinity ligand, 77.2%; low-affinity ligand, 22.8%) (GC1118 vs. cetuximab, *p* = 0.09; Figure 3A, upper panel and Appendix A). Similarly, treatment with GC1118 or cetuximab significantly inhibited CRC-077T growth (*KRAS*-wild-type; high-affinity ligand, 51.6%; low-affinity ligand, 48.4%) with TGII values of 6.8% (*p* = 0.0016) and 10.4% (*p* = 0.0015), respectively (Figure 3A, upper panel and Appendix A), suggesting comparable antitumor potency of GC1118 to cetuximab in patients harboring wild-type *KRAS*. Interestingly, GC1118 showed a significantly superior efficacy (TGII: 36.7%, *p* = 0.006) to cetuximab (TGII: 36.7%, *p* = 0.006) in CRC-001T PDXs (*KRAS*-wild-type; high-affinity ligand, 88.7%; low-affinity ligand, 11.3%) (GC1118 vs. cetuximab, *p* < 0.001; Figure 3A, upper panel and Appendix A).

Of note, GC1118 showed a more significant inhibitory effect on tumor growth than did cetuximab in cases of *KRAS*-mutant CRC (Figure 3A, lower panel and Appendix A). In CRC-026T PDXs (*KRAS* G12D; high-affinity ligand, 84.2%; low-affinity ligand, 15.8%), TGII values for GC1118 and cetuximab were 47.9% (*p* = 0.006) and 97.5% (*p *= 0.053), respectively (*p* = 0.001; Figure 3A, lower panel and Appendix A). Further, TGII values for GC1118 and cetuximab in CRC-034T (*KRAS* G12V; high-affinity ligand expression, 72.2%; low-affinity, 27.8%) were 34.5% (*p* = 0.023) and 103.6% (*p* = 0.12), respectively (GC1118 vs. cetuximab, *p* = 0.019; Figure 3A, lower panel and Appendix A). Finally, treating CRC-088T PDXs (*KRAS* G12V; high-affinity ligand, 61.3%; low-affinity ligand, 38.7%) with GC1118 and cetuximab resulted in TGII values of 10.8% (*p* = 0.001) and 47.6% (*p* = 0.91), respectively (GC1118 vs. cetuximab, *p* = 0.012; Figure 3A, lower panel and Appendix A). Overall, no significant differences were observed in the body weights of animals over the course of this study (Appendix A). The potent inhibitory effect of GC1118 on high-affinity EGFR ligand-induced signaling is more pronounced for downstream signaling molecules including AKT and ERK1/2 [8]. GC1118 and cetuximab resulted in variable inhibitory effects on ERK and AKT activation compared to that in the control group according to each PDX, as measured by IHC (Figure 3B,C and Appendix A) and immunoblotting (Appendix A**)**. Overall, ERK and AKT signaling activities were significantly suppressed after treatment with GC1118 alone compared to that with cetuximab alone.

### 2.3. Activation of AKT Signaling Confers Resistance to GC1118 Monotherapy in KRAS-Mutant CRC PDX Models

The combined TGII from a panel of CRC PDXs revealed that GC1118 treatment inhibited tumor growth significantly better than cetuximab in *KRAS*-mutants (Figure 3 and Appendix A); however, complete tumor regression was not observed. In seven CRC PDXs with varying levels of basal EGFR, AKT, and ERK1/2 activation before GC1118 treatment (Figure 4A), including an additional CRC-024T model (*KRAS* G12D; high-affinity ligand, 88.8%; low-affinity ligand, 11.2%; high basal AKT activity) with resistance to GC1118 and cetuximab (TGII-GC1118 = 65.6) (Appendix A), the efficacy of GC1118 (TGII) showed a significant positive correlation with basal AKT activity only (Pearson’s r = 0.82, *p* = 0.024) (Figure 4B).

PI3K activity is the main predictor of mitogen-activated protein kinase kinase (MEK)-inhibitor resistance in *KRAS*-driven CRC [33,34] and thus, the additional use of a PI3K inhibitor could overcome resistance to MEK inhibition [35]. Although KRAS can directly activate PI3K signaling by binding to the p110-PI3K subunit, there is increasing evidence that PI3K activation, following MEK inhibition, is correlated with RTK activity, providing the foundation for the use of RTK inhibitors in *KRAS*-mutant CRC [36]. Based on these findings, we performed preliminary in vivo experiments, evaluating the combination of GC1118 and the dual PI3K/mTOR inhibitor BEZ-235 [27], in a relatively GC1118-resistant CRC-024T model (KRASG12D showing high basal AKT activity (Figure 5). Here, cetuximab was inactive (TGII = 109.4%, *p* = 0.600), whereas GC1118 (TGII = 65.6%, *p* = 0.255) or BEZ-235 (TGII = 67.4%, *p* = 0.103) alone had moderate antitumor effects (Figure 5A and Appendix A). Furthermore, the combination of the two molecules exerted significant inhibitory effects on tumor growth (TGII = 31.6%; *p* = 0.007; Figure 5A) with no reduction in body weight (Figure 5B) and without any other signs. We also confirmed significant inhibitory effects on AKT and ERK1/2 activity using IHC (Figure 5C,D and Appendix A) and immunoblotting (Figure 5E).

## 3. Discussion

As CRCs differ in clinical presentation, molecular heterogeneity, and the involvement of several molecular pathways and molecular changes [5,37], PDXs represent the fastest and most effective approach to uncover active therapeutic agents for CRC [24,25,26]. In contrast to previous studies, we utilized the PDX platform to evaluate the efficacy of GC1118 and its mechanism of action, as the induction and expression of high-affinity EGFR ligands have been reported to be more prevalent in CRC tumor xenografts than in in vitro cultures [8]. GC1118 is a human anti-EGFR IgG1 antibody that differs from existing anti-EGFR MoAbs, such as cetuximab and panitumumab, in its constant region, affinity, mode of action, and efficacy [8,20], exhibiting superior binding affinity (resulting in ADCC) to both the low- and high-affinity variants of FcγRIIIa compared to cetuximab [8,20]. Moreover, the use of Bagg albino (BALB)/c nude mice with intact innate immune systems could allow for the evaluation of GC1118-mediated ADCC through Fc receptors present on immune effector cells such as macrophages, monocytes, and natural killer cells [8,11,38].

A subset of CRCs lacking *KRAS* pathway mutations and showing “EGFR addiction” is treatable using two EGFR-targeting MoAbs, namely cetuximab and panitumumab [4,9]. When the oncogenic stimulus occurs downstream, such as in tumors with *KRAS* mutations, resistance to these therapies arises [4,5,7,12,16,39,40]. *KRAS* mutations in CRC are associated with a more rapid onset and aggressive metastasis, making it clinically more challenging [16,41,42]. Herein, we showed that efficiently blocking high-affinity EGFR ligands with GC1118 induces superior therapeutic benefits in *KRAS* mutated CRC PDX platform refractory to cetuximab. In addition, the basal up-regulated AKT pathway was correlated with lower efficacy of GC1118, and our preliminary, promising results indicated that GC1118 combined with the PI3K/mTOR/AKT inhibitor BEZ-235 showed improved antitumor effects on *KRAS*-mutant tumors with intrinsically high AKT activity with favorable safety, encouraging further studies using novel therapeutic combinations to treat clinically-aggressive *KRAS*-mutant CRC showing elevated ratios of high- to low-affinity EGFR ligands and PI3K/mTOR/AKT signaling (Figure 6).

Constitutively active MAPK signaling in *KRAS*-mutated CRC promotes epithelial–mesenchymal transition and cancer stemness, independent of external EGFR stimulation [43,44]. Further, persistent downstream signaling through the RAS axis due to *KRAS* mutations can activate multiple processes involved in tumor progression and metastasis without the influence of EGFR and other cell surface receptor kinases. Previous studies have reported a significant association between EREG/AREG expression and cetuximab response in *KRAS*-wild-type patients, but not in *KRAS*-mutant patients [6,8,9,10,20,21,45,46]. Therefore, there is an unmet need for novel EGFR-targeting therapies as alternative treatment options. Our results showed that CRC PDXs harboring *KRAS* mutations expressed remarkably higher levels of high-affinity EGFR ligands than *KRAS*-wild-type tumors, suggesting that the expression levels of EGFR ligands could be used as biomarkers to predict the therapeutic response to EGFR-targeting strategies. Although EREG and AREG are predominant EGFR ligands expressed in CRC, and only a small fraction of high-affinity ligands is expressed [29], upon downstream activation of the EGFR/RAS/MAPK axis owing to a mutated KRAS effector, the expression of AREG and EREG ligands would be biologically irrelevant in terms of any benefit from cetuximab [8,20,21,45]. The observed superior antitumor potency of GC1118 over cetuximab against CRC PDXs harboring activating *KRAS* mutations could be due to the strong inhibitory activity of the interaction between EGFR and high-affinity EGFR ligands [8,20,21], providing a rationale for clinical application of the expression pattern of EGFR ligands as a novel biomarker predictive of the response to GC1118 in treating patients with refractory mCRC. Supporting our work, increased secretion of the high-affinity EGFR ligands TGF-α and BTC by some *KRAS*-mutant clones has been suggested to be a paracrine resistance mechanism to anti-EGFR antibodies in CRC models [47,48,49]. Considering the significant roles of high-affinity EGFR ligands in modulating the tumor microenvironment and inducing resistance to various cancer therapeutics, our study suggests potential therapeutic advantages for GC1118 in terms of efficacy and the range of patients for whom it will be beneficial. Genetic and molecular mechanisms determining the ratio of high-affinity/low-affinity EGFR ligands, other than *KRAS* mutation status, should be elucidated through further comparative analyses of the therapeutic effects of GC1118 on CRC PDXs secreting mainly high- or low-affinity EGFR ligands using a larger panel of heterogenous CRC PDXs.

Here, importantly, we found that resistance to GC1118 was associated with increased activation of AKT signaling, suggesting that persistent activation of the PI3K/AKT/mTOR signaling axis by high-affinity EGFR ligands could be a potential feedback and resistance mechanism inducing EGFR inhibition. Although we focused on CRC PDX cases harboring only KRAS mutations to validate the potential of combined PI3K/mTOR/AKT and EGFR inhibition in *KRAS*-mutant CRC cells with high AKT activity due to several mechanisms such as the ratio of high- to low-affinity EGFR ligands, further investigations on CRC PDXs harboring concurrent mutations in both *KRAS* and the genes activating PI3K/mTOR/AKT pathway (e.g., PIK3CA) are required to strengthen the importance of PI3K/mTOR/AKT pathway in the resistance to GC-1118. Genetic mutations in the PI3K and MAPK pathways are frequently implicated in CRC [10,11,12]. CRC patients with PIK3CA and KRAS mutations are unlikely to respond to the inhibition of the MEK pathway alone or the PI3K pathway alone but will require effective inhibition of both MEK and PI3K/AKT signaling pathways [12,13,16,34,39,50,51,52,53,54,55]. For example, BEZ-235, in combination with EGFR inhibitors, is more effective for less mTOR inhibitor-sensitive and EGFR inhibitor-resistant CRC cell lines, especially HCT116 (which harbors *KRAS* and *PIK3CA* mutations), as shown in a recent study [39]. Previous findings suggest that acquired resistance to anti-EGFR MoAbs biochemically converges on RAS/RAF/MEK/ERK and PI3K/mTOR/AKT pathways, coupled with cross-talk mechanisms between other members of the EGFR family, such as HER2 and HER3, as well as IGF1R [39,55,56,57,58,59]. Additionally, it is well established that autophagy is associated with resistance to anti-EGFR MoAb therapy because EGFR stimulates multiple downstream signaling pathways that affect autophagy, including the PI3K–AKT–mTOR axis [7,60]. Combination therapy comprising anti-EGFR MoAbs together with autophagy-inducing PI3K/mTOR inhibitors could be used to develop an active therapeutic strategy for mCRC patients by inducing autophagic cell death [61,62].

Activating mutations in PIK3CA (phosphatidylinositol-4,5-bisphosphate 3-kinase, catalytic subunit alpha) are present in 15%–20% of CRCs, and the prevalence of PIK3CA exon 9 and/or exon 20 hotspot mutations increases continuously from rectal (10%) to cecal (25%) cancers, supporting the colorectal continuum paradigm [13,14,55,63,64,65,66,67,68,69,70,71,72,73]. Coexisting *PIK3CA* and *KRAS* mutations, which occur in approximately 8%–9% of CRC cases [55,66,67,68,73,74,75,76,77,78], predict resistance to anti-EGFR therapy, as well as worse prognosis, in CRC [16,39,52,55,66,68,76,79,80,81,82,83,84,85,86]. Interestingly, mutations in *PIK3CA* exon 9 (and to a lesser extent exon 20) are associated with features of the traditional serrated pathway (CpG island methylator phenotype-low (CIMP-low)/*KRAS* mutation) of tumorigenesis [66,68,76,78]. Insight into KRAS-driven CRCs will stimulate new research to find the best approach to treat this aggressive type of cancer, encouraging further evaluations of novel combination strategies including PI3K/mTOR/AKT inhibitors [39,56,87]. Although only one case was tested in the present study, our data highlight the potential of combined PI3K/mTOR and EGFR inhibition for *KRAS*-mutant CRC cells with relatively high levels of high-affinity EGFR ligands, although further investigation on the therapeutic efficacy, mode of action, and tolerability of this combination based on additional *KRAS*-mutant PDX models concurrently harboring other genetic alterations (with different genetic backgrounds) is required. In fact, there were three cases with mutations in both KRAS and PIK3CA among our panel (CRC-017T: KRAS G13D, PIK3A Q546K, TP53 R81X and P27R; CRC-021T: KRAS G13D); however, they could not be used for in vivo validation due to the difficulty in obtaining sufficient PDX cells for in vivo combination efficacy test. The verification of the synergy of GC1118 and BEZ-235 in several *KRAS*-mutant CRC PDX cases less susceptible to GC1118 by high AKT activity is essential to provide clinical reliability and strong support for our hypothesis, highlighting the potential of combined PI3K/mTOR and EGFR inhibition in *KRAS*-mutant CRC cells with relative high levels of high-affinity EGFR ligands. Our data highlight the potential of combined PI3K/mTOR/AKT and EGFR inhibition in *KRAS*-mutant CRC cells with relatively high levels of high-affinity EGFR ligands, with a need for further investigations on the therapeutic efficacy, mode of action, and tolerability for optimizing this combination in additional *KRAS*-mutant PDX models concurrently harboring other genetic alterations. As the low frequency of these double-mutant cases underscores the need for collaborative international efforts to undertake such drug combination studies, optimizing the design of such clinical trials for CRC requires a detailed knowledge of the prevalence of these respective mutant genotypes.

In summary, the superior inhibitory activity of GC1118 on high-affinity EGFR ligands, for which current clinical antibodies show restricted inhibitory activity, reflects the potential therapeutic advantage of this drug for the treatment of cancer in which high-affinity EGFR ligands are implicated in tumor progression, metastasis, and resistance to current cancer therapeutics. Although future work should focus on the development of predictive biomarkers and hypothesis-driven rational combinations, GC1118 might be of therapeutic benefit, alone or in combination with other agents, for *KRAS*-mutant mCRCs with elevated ratios of high- to low-affinity EGFR ligands and intrinsic PI3K–AKT pathway activation. Further validation based on mouse trials is required based on an expanded CRC PDX panel to overcome the heterogeneity encountered in the clinic and optimize clinical trial designs and further define a patient enrichment strategy.

## 4. Materials and Methods

### 4.1. CRC Patient Clinical Information

All CRC patients provided informed consent for the use of their tissues in this study, in accordance with protocols approved by the Samsung Medical Center (Seoul, Korea) Institutional Review Boards (IRB 2010-04-004). Sequencing analysis (after polymerase chain reaction (PCR) amplification) was performed on 51 patient-derived tissues to confirm the presence of *KRAS*, *BRAF*, *PIK3CA*, and *TP53* mutations. Clinical information derived from histological examination and diagnosis based on biopsies from 51 patients with CRC was provided by the Samsung Medical Center. PCRs were carried out in a 20 μL reaction volume containing 100 ng genomic DNA, 10 pmol of each primer, and Maxime PCR premix (iNtRON Biotechnology, Seongnam, Korea). Bidirectional sequencing was performed using a BigDye Terminator v1.1 kit (Applied Biosystems, Foster City, CA, USA) on an ABI 3130XL Genetic Analyzer (Applied Biosystems). Sequence analysis was performed using the software package Sequencher 4.10.1 (Gene Codes Corporation, Ann Arbor, MI, USA).

### 4.2. Establishment of CRC PDXs and Analysis of EGFR Ligand Expression

To evaluate autocrine-derived EGFR ligands (and not paracrine ligands produced by stromal cells in the tumor microenvironment), we implanted CRC tumor fragments obtained from 51 patients into the subcutaneous layer of immunodeficient BALB/c nude mice, generating PDXs, as described previously [88]. Animal experiments were conducted in accordance with the Institute for Laboratory Animal Research Guide for the Care and Use of Laboratory Animals, and all protocols were approved by the Samsung Medical Center. Tumors that reached a volume of 1000 mm^3^ were considered tumorigenic. Tumor tissues were isolated from subcutaneous CRC PDXs when the tumor volume reached approximately 200 mm^3^. The tumors were homogenized, extracted in 1 mL lysis buffer (Cell Signaling Technology, Danvers, MA, USA), and centrifuged to remove tissue residue. The supernatant components were measured using multiplex ELISA arrays. Human EGF/HB-EGF/TGF-α/BTC/AREG ELISA kits (Ray Biotech, Norcross, GA, USA) and human EREG ELISA kits (USCN Life Science Inc., Houston, TX, USA) were used according to the manufacturers’ protocols to quantify the expression level of each EGFR ligand.

### 4.3. In Vivo Therapeutic Efficacy Evaluation Using a Panel of CRC PDX Models

All in vivo experiments were conducted according to the guidelines of the Association for Assessment and Accreditation of Laboratory Animal Care, from the Samsung Medical Center Animal Use and Care Committee (Approval No. 20151209001) and the National Institute of Health (NIH; Bethesda, MD, USA) Guide for the Care and Use of Laboratory Animals (NIH publication 80-23). We propagated CRC PDXs to evaluate the therapeutic efficacy of cetuximab, GC1118, and BEZ-235 by implanting PDX tumors into the flanks of 6–8-week-old female BALB/c nude mice purchased from Orient Bio Inc. (Seongnam, Korea). Tumors were harvested when they reached approximately 500 mm^3^, and dissociated single cells were isolated, added to Hank’s Buffered Salt Solution medium and Matrigel Basement Membrane Matrix mixture (1:1), and subcutaneously injected into the flanks of 6–8-week-old female BALB/c-nude mice.

When tumors reached approximately 200–250 mm^3^, the animals were randomized into groups based on tumor volume to minimize intragroup and intergroup variation (*n* = 3–7 mice/group). The start of dosing was defined as day 1, and tumor volumes and body weights were measured twice per week for 28–52 days, depending on the growth of each PDX. Tumor volume was calculated as (length × width^2^) × 0.52 [32]. Relative tumor volume was normalized to the initial tumor volume on day 1. GC1118 or cetuximab was administered at 50 mg/kg (1 mg/mouse) [8,20]. A vehicle was administered intraperitoneally twice per week, and BEZ-235 was administered at 20 mg/kg (0.4 mg/mouse) orally five times per week [89]. TGII values was used for antitumor efficacy [32]. Mice were monitored daily for signs of toxicity. After sacrificing each mouse, tumor tissue was harvested and divided into two parts, one for IHC examination and the other for protein extraction.

### 4.4. IHC

At the indicated post-treatment times, additional tumor-bearing mice were sacrificed and tumors were harvested to generate formalin-fixed paraffin-embedded (FFPE) specimens. FFPE samples were processed according to conventional experimental protocols for IHC analysis. Specimens were fixed with 4% paraformaldehyde in phosphate-buffered saline (PBS; Gibco), embedded in paraffin, and cut into sections that were blocked and permeabilized with 0.3% triton X-100 (Sigma-Aldrich, St. Louis, MO, USA) and 10% horse serum in PBS. Deparaffinization and antigen retrieval were followed by primary antibody staining and hematoxylin counterstaining. Primary antibodies used to label proteins were as follows: anti-phospho-AKT (Ser473; 1:50) and anti-phospho-ERK1/2 (Thr202/Tyr204; 1:100) (Cell Signaling Technology). These were labeled with secondary antibodies, as previously described [90]. To quantify AKT and ERK activity based on IHC, images were captured with an automatic histologic imaging system (TissueFAXS, TissueGnostics GmbH, Vienna, Austria). The expression of anti-phospho-AKT and anti-phospho-ERK1/2 was quantified by HistoQuest Analysis Software using TissueFAXS system (TissueGnostics) after defining regions of interest. Several parameters, such as nuclei size and intensity of staining, were adjusted to achieve optimal cell detection. Cells were plotted to scattergrams according to human-specific marker signals. Cutoff thresholds were determined using signal intensity of the secondary antibody alone as a negative control. Positive cell counts from images of immune-histolabeled sections were measured by two independent observers blinded to the experimental conditions. Mean values for positive cells counted in five locations were evaluated.

### 4.5. Immunoblotting Analysis

The tissues of tumor-bearing mice treated with GC1118, cetuximab, or BEZ-235 were prepared for western blotting. All tissues were lysed in NP40 buffer (50 mM Tris, pH 7.4, 250 mM NaCl, 5 mM EDTA, 50 mM NaF, 1 mM Na_3_VO_4_, 1% Nonidet P-40, 0.02% NaN_3_) with additional protease inhibitor cocktail tablets (Sigma-Aldrich, St. Louis, MO, USA) and phenylmethanesulfonyl fluoride (Sigma-Aldrich). Equal amounts of protein were subjected to SDS-PAGE and transferred to polyvinylidene difluoride membranes (Millipore). After blocking nonspecific binding with 5% skimmed milk or 5% bovine serum albumin (BSA)(Sigma-Aldrich) for 2 h at room temperature, the membranes were incubated with the indicated primary antibodies overnight at 4 °C and then with the appropriate secondary antibodies for 1 h at room temperature. EGFR-mediated downstream pathway proteins were confirmed using rabbit monoclonal antibodies, including anti-phospho-EGFR, anti-EGFR, anti-phospho-AKT(Ser473), anti-AKT, anti-phospho-ERK1/2 (Thr202/Tyr204), anti-ERK1/2 (Cell Signaling Technology Danvers, MA, USA), and anti-β-actin (Abcam, Cambridge, MA, USA) antibodies, with the Amersham ECL Prime western blotting detection reagent (GE Healthcare, Anaheim, CA, USA). For quantification, images were acquired and the signal intensity of each protein band was quantified using ImageJ software (NIH, Bethesda, MD, USA) and normalized to β-actin. The activities of EGFR, AKT, and ERK1/2 were determined by normalization with their total pairs, namely pEGFR/EGFR, pAKT/AKT, and pERK1/2/ERK1/2, respectively [91].

### 4.6. Statistics

Results were analyzed for statistical significance using GraphPad Prism V5.04 software and SPSS v.16 (SPSS Inc., Chicago, IL, USA). All data are expressed as the mean ± standard error of the mean (SEM) from at least three independent experiments. Two-tailed t-tests and one-way analysis of covariance were used to assess the differences between two groups of continuous variables, and *p* values < 0.05 were considered significant. Pearson’s correlation coefficients and two-tailed significance were calculated for each case. An unpaired t-test was used to compare TGIIs between different treatments. We used a key to indicate levels of significance as follows: * *p* < 0.05, ** *p* < 0.01, and *** *p* < 0.001.

## Figures and Tables

**Figure 1 ijms-20-05894-f001:**
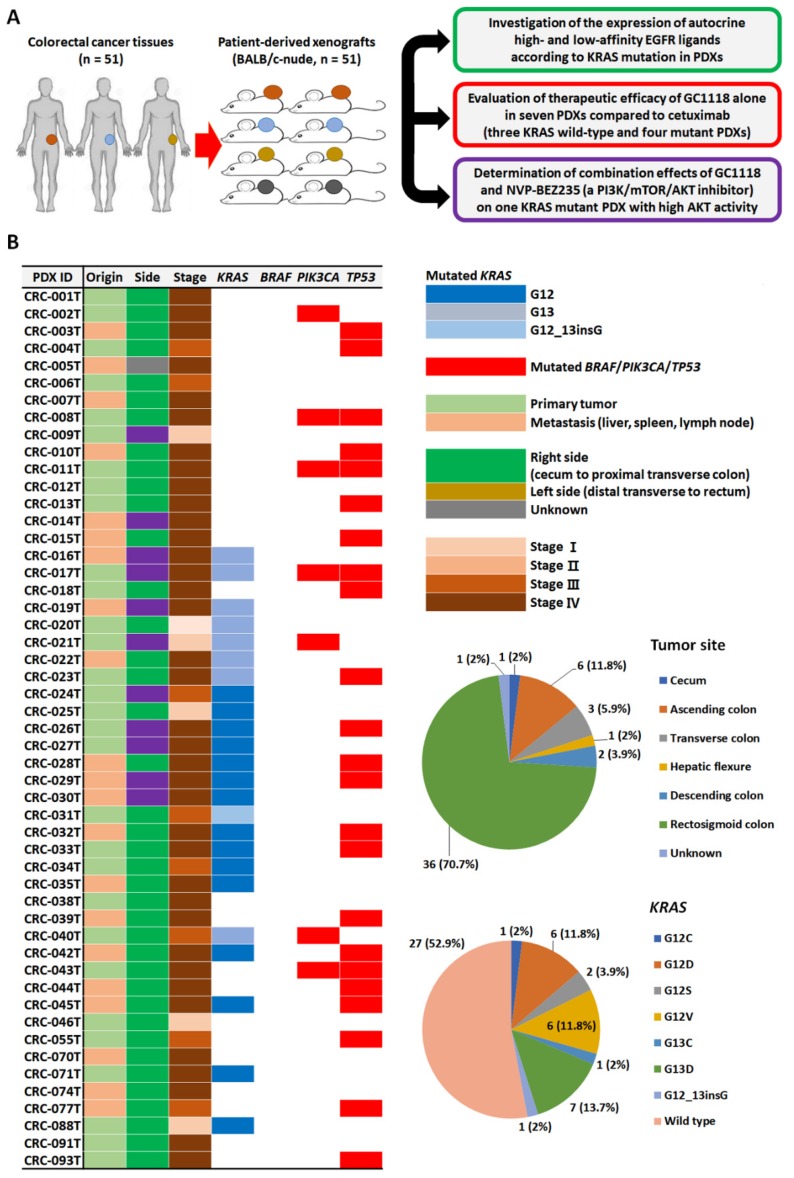
Genomic characterization of colorectal cancer (CRC) patient-derived xenografts (PDXs) and expression of epidermal growth factor receptor (EGFR) ligands. (**A**) Schematic overview of the analytical workflow used in the study to evaluate therapeutic efficacy. (**B**) A characteristic profile of the associated clinical information and genetic abnormalities of 51 patients with CRC (left panel) and distribution of cases according to the tumor site (right panel, upper) and prevalence of KRAS proto-oncogene GTPase (*KRAS*) mutations (right panel, lower) in our CRC PDX panel.

**Figure 2 ijms-20-05894-f002:**
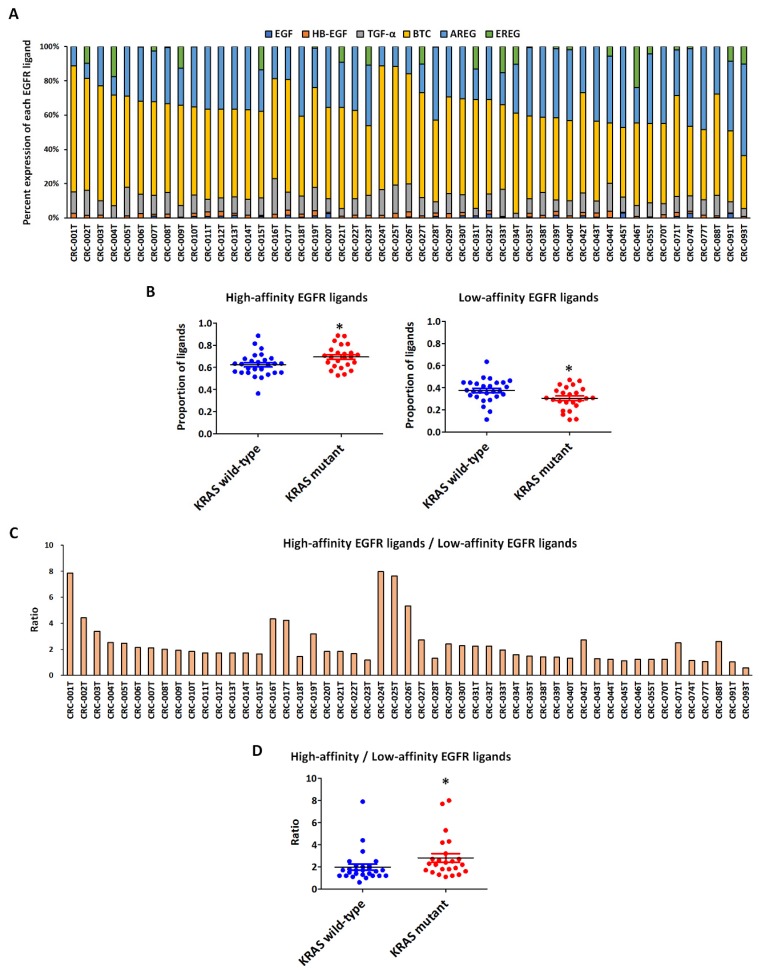
Percent distribution of ligand expression levels in 51 colorectal cancer (CRC) patient-derived xenografts (PDXs). (**A**) Percent ligand expression levels for EGF, HB-EGF, TGF-α, BTC, AREG epidermal growth factor (EGF), heparin-binding EGF-like growth factor (HB-EGF), transforming growth factor α (TGF-α), betacellulin (BTC), amphiregulin (AREG) and epiregulin (EREG) in 51 individual CRC PDXs as determined by ELISA. (**B**) Proportion of high- and low-affinity EGFR ligands in CRC PDX models according to their *KRAS* status. The graph shows the mean and standard error of the mean (SEM). * *p* < 0.05. (**C**) High/low-affinity ligand expression ratios in 51 individual CRC PDX models. (**D**) High/low-affinity ligand ratio in CRC PDX models according to their *KRAS* status. The graph shows the mean and SEM. * *p* < 0.05.

**Figure 3 ijms-20-05894-f003:**
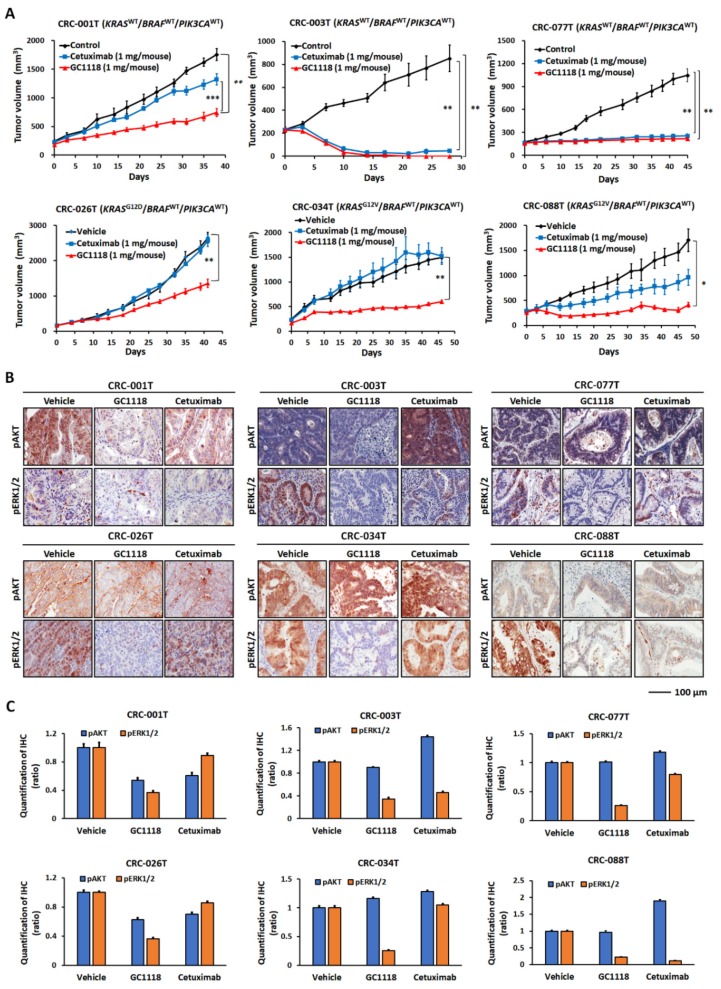
Antitumor activity of GC1118 against both *KRAS*-wild-type and mutant colorectal cancer (CRC) patient-derived xenografts (PDXs). (**A**) Tumor growth after GC1118 or cetuximab treatment in six individual CRC PDXs. * *p* < 0.05, ** *p* < 0.01, *** *p* < 0.001. (**B**) Representative images of immunohistochemistry (IHC) detection of AKT and ERK signaling activity in six individual CRC PDXs. (**C**) Quantification of AKT and ERK activity, as measured by IHC. The results in the bar graph are shown as the means and standard error of means (SEM). Statistical significance is summarized in Appendix A.

**Figure 4 ijms-20-05894-f004:**
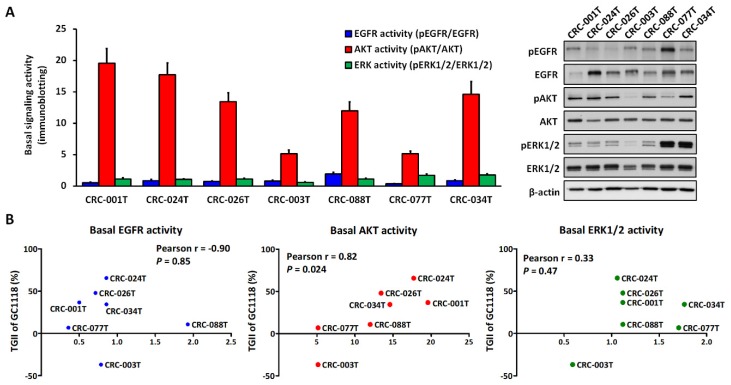
Correlation analysis of the inhibitory effects of GC1118 and basal signaling activity level in a colorectal cancer (CRC) patient-derived xenograft (PDX) model. (**A**) Basal activity levels of EGFR, AKT, and ERK1/2 pathways based on western blotting using tumor xenografts from each CRC PDX model. The tumor samples were isolated when the tumor xenografts reached 200 mm^3^. For quantification, images were acquired and signal intensity of each protein band was quantified using the ImageJ software (NIH, Bethesda, MD, USA) and normalized to β-actin. The activities of EGFR, AKT, and ERK1/2 were determined by normalization with their total pairs, namely pEGFR/EGFR, pAKT/AKT, and pERK1/2/ERK1/2, respectively (**B**) Pearson’s correlation analysis was performed to analyze the correlation between EGFR, AKT, and ERK1/2 activities (X-axis) and the tumor growth inhibition index (TGII, Y-axis) in six CRC PDXs.

**Figure 5 ijms-20-05894-f005:**
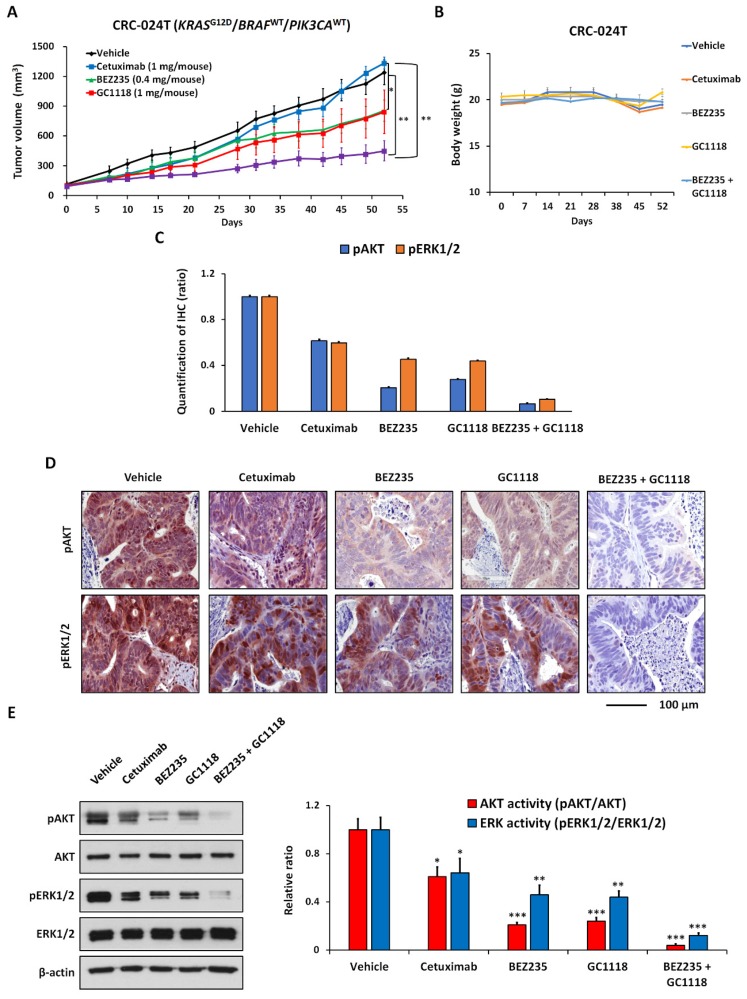
Antitumor activity of GC1118 and BEZ-235 in the colorectal cancer (CRC)-024T patient-derived xenograft (PDX) model. (**A**) Tumor growth after individual or combined GC1118 and BEZ-235 treatment in the CRC-024T PDX model. The results in the graph are shown as means and standard errors of means (SEM). * *p* < 0.05, ** *p* < 0.01. (**B**) Mouse body weight during the course of the in vivo study at indicated time points. Error bars represent SEM. (**C**) Analysis of signaling pathways based on immunohistochemistry (IHC) detection of AKT and ERK1/2 signaling activities in the CRC-024T PDX model after treatment with GC1118 and BEZ-235. The results in the graph are shown as the mean and SEM. The values indicating the statistical significance between each group based on a T-test were described in Appendix A. (**D**) Representative IHC images of AKT and ERK1/2 signaling activities in the CRC-024T PDX model treated with GC1118 and BEZ-235. (**E**) Analysis of signaling pathways by immunoblotting for AKT and ERK1/2 signaling activities in the CRC-024T PDX model treated with GC1118 and BEZ-235. For quantification, images were acquired and signal intensity of each protein band was quantified using the ImageJ software (NIH, Bethesda, MD, USA) and normalized to β-actin. The activities of EGFR, AKT, and ERK1/2 were determined by normalization with their total pairs, namely phospho-EGFR/EGFR, phospho-AKT/AKT, and phospho-ERK1/2/ERK1/2, respectively. The results in the graph are shown as SEM. The significant difference between vehicle and each treatment group is indicated. * *p* < 0.05, ** *p* < 0.01, *** *p* < 0.001.

**Figure 6 ijms-20-05894-f006:**
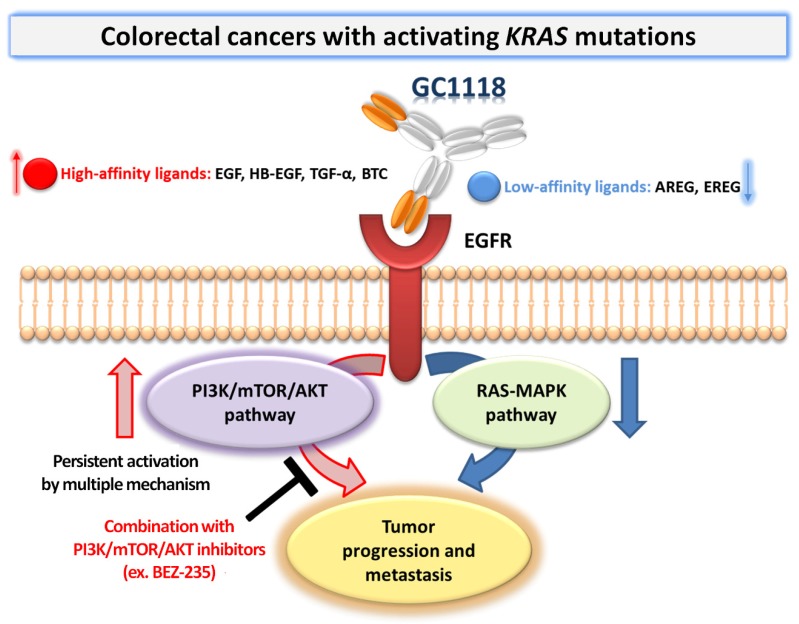
Our hypothesis of the mechanisms underlying the inhibitory effects of GC1118 on colorectal cancer (CRC) with *KRAS* mutations and circumventing resistance to GC1118 by combining this drug with PI3K/mTOR/AKT inhibitors (BEZ-235 in this study) in *KRAS*-mutated CRCs with persistently activated PI3K/mTOR/AKT signaling.

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
