# Peer review of "Promising Therapeutic Efficacy of GC1118, an Anti-EGFR Antibody, against KRAS Mutation-Driven Colorectal Cancer Patient-Derived Xenografts"

_ijms, 2019, doi:10.3390/ijms20235894_

Round 1
Reviewer 1 Report
The authors provide very comprehensive analysis of EGFR inhibition inn PDX systems which will be important for clinical interventions.
Minor comments:
mentions which statistics was used to generated p-values
how does the authors quantify akt or erk activity? this can be explained.
figure 5 did the authors check for ki67 and such after treatment for
decreasing tumor size?
Reviewer 2 Report
This is an original study by Hye Won Lee et al., that explores the anti-tumor efficacy of the novel anti-EGFR moAb, namely GC1118, against wild type and KRAS-mutation driven CRC xenografts in a mouse model. Although, it is an interesting study with immense translational potential in cancer immunotherapeutic era, the manuscript overall suffers from structural weaknesses and incomplete data presentation.
Specifically:
Major comments
1) The numbering of the panels in Figures 1, 2, 3 and 4 are misleading and not consistent within the text and the figure legends. The use of capital or small letters for the labeling of the figure panels should be consistent.
2) Most of the results sections have not been structured properly. For example section 2.1 starts with Figure 1, without including analysis of part of the data in the text, but only in the Figure legend. All figure numbers and panels should be cited within the data description and not standing alone, to easy the reading
3) Section 2.2.: The authors mention that they included in their vivo studies 3 KRAS w/t and 4 KRAS mutant PDXs. However, in the data presented in the associated Figure 3 (all panels) one mutant is missing (CRC-024T). Furthermore, no description of this sample is included in the data analysis (section 2.2). Thus, it is not clear whether the data derived by this sample have been included in the overall statistical analysis or not.
4) Section 2.2 and Figure 3: Given the available samples, the authors have used relatively small sample number for analyzing and comparing the responses of KRAS W/T (N=3) and mutant (N=3) to GC1118 and cetuximab. Therefore, it is difficult to come to a statistically significant and safe conclusion about a more potent therapeutic efficacy of GC1118 over cetuximab, especially in KRAS W/T samples, as the authors claim in the title of section 2.2. Characteristically, Figure 3A shows that only 1 out of 3 KRAS W/T samples had better response in GC1118 treatment, compared to cetuximab. As such the obtained findings have to be validated in additional samples.
5) Section 2.2, line 147: The authors mention that all PDXs used in Figure 3 were PIK3CA wild type. It would be interesting the authors to explore the efficacy of GC1118 on PIK3CA mutant PDXs, in addition to KRAS status. The obtained findings may unreveal any putative impact of GC1118 on PI3K constitutive signaling.
6) In Figure 1B, the authors show that KRAS mutant PDXs produce more high affinity (HA) EGFR-ligands, while the W/T PDXs mainly secrete low affinity (LA) EGFR ligands. Although it is reasonable the authors to select HA EGFR-ligand-expressing KRAS W/T and mutant PDXs samples for their further analysis, it would be interesting to test any putative statistically significant differences in GC1118-mediated responses between PDXs expressing mainly HA- or LA-EGFR ligands.
7) Figure 4: Panel labeling is misleading in both figure legend and section 2.3 text (1st paragraph). It’s not clear by the description, whether Figure 4 panels refer to basal activities of the indicated gene products or activities after PDX treatment with GC1118.
8) Figure 5: Sample number is extremely small (N=1) for safe conclusions about efficient synergy of GC1118 and BEZ-235 in CRC PDXs. Misleading panel labeling. More samples should be included in the analysis
9) Figure 6 should be cited and discussed in discussion not in Figure 5 legend.
Minor comments
1) Supplemental Table 2: What do the numbers in the last two raws of the table, represent? Table legend should be more detailed and specific. CRC033T should be CRC003T.
2) Figure 3C: SEMs should be included in the graphs
3) Line 84-88: Rephrase sentence. One gets the false impression that activation of PTEN pathway is involved in the activation of EGFR-independent downstream signaling cascades involved in CRC resistance to therapy by anti-EGFR MoAbs.
4) Line 103: ‘characteristics’ should be ‘characteristic’
5) Lines 191-193: Sentence should be rephrased
6) Figure 4 1st graph: SEMs are missing. What ratio does the Y axis represent?
7) Figure 5C (left panel) and 5E (right panel): SEMs are missing
8) Line 219: Add ‘or combination’ after ‘treatment’
9) Lines 195-198: Rephrase sentence
10) Lines 210-215: Rephrase sentence
11) Editing of English language and style are required, especially in figure legends.
Round 2
Reviewer 1 Report
The authors have satisfactorily responded. I recommend Accepting the manuscripts.
Reviewer 2 Report
The authors have put significant effort to successfully revise their ms and correct most of its structural weaknesses. However, there is still space for further improvement. Also the ms (especially the new additions) needs further English language editing.
I’ll comment on authors’ responses given for the major points raised in my first review:
1 and 2) Satisfactory revision. Authors should again be extremely careful with the use of small, or capital letters for the labeling of figure panels. They have to choose either all capital, or all small.
3) The confusion still exists. Since growth data are not provided, it will better the authors to exclude sample CRC-024T for the analysis shown in Figure 3 and state in the text (section 2.2) that they worked only with six CRC PDXs. Sample CRC-024T and its specific characteristics can be introduced in section 2.3 and Figures 4 and 5. Lines 150-166 need extensive revision for clarity.
4) From the data the authors present, I am still not convinced about a statistically significant predominance of GC1118 response over cetuximab, in KRAS W/T samples (Figure 3A, N=3, better GC1118 efficacy only in 1 out of 3). If the authors don’t have the means to provide additional data derived from a larger number of KRAS W/T samples, then it will better to focus their conclusions on the better efficacy of GC1118 over cetuximab ONLY in KRAS mutant PDXs. As such, any relevant statements should be avoided within the text and the title of 2.2 section should be altered accordingly.
5) I suggest any explanations and relevant references given about the lack of the proposed (by the reviewer) studies to be also included in the general discussion section. The authors also mention in their response that ‘Unfortunately, the cases with the mutations in both KRAS and PIK3CA for in vivo validation was not available among our panel of CRC PDXs’’. However in the sample panel used for the study, there were 3 cases of both KRAS and PIK3CA mutants…
6) Similarly, the discussion about the raised question and its significance should also be included in the discussion section
7) Satisfactory revision, further improvement still needed. Figure 4A shows fold change of expression relative to sample CRC-001T. Why the CRC-001T is used as control? Figure 1A should present the actual expressions derived from WB relative to actin which serves as internal control.
8) Given that the sample number is extremely small (N=1) and the authors don’t have the means to include additional samples in the study, any conclusion about a statistically significant synergistic effect GC1118 and NVP-BEZ235 on resistant CRC tumors, should be avoided. Instead they can refer to evidences derived from their preliminary findings. As such relevant statements within the text and the title of section 2.3 should be revised, accordingly.
9) OK
Response to reviewer 2
Response to comments from anonymous referee #2 General Remarks: The authors have put significant effort to successfully revise their ms and correct most of its structural weaknesses. However, there is still space for further improvement. Also the ms (especially the new additions) needs further English language editing. I’ll comment on authors’ responses given for the major points raised in my first review. Response: We greatly thank the reviewer for providing additional useful comments on our firstly revised manuscript.
Major comment 1. 1 and 2) Satisfactory revision. Authors should again be extremely careful with the use of small, or capital letters for the labeling of figure panels. They have to choose either all
capital, or all small. Response: We really appreciate the reviewer’s previous comments on figure citations and labeling of figure panels for clear description of our findings. Per suggestion, during resubmission of second round revision, we rechecked to correct any additional errors in the figure panels, text and figure legends.
2. 3): The confusion still exists. Since growth data are not provided, it will better the authors to exclude sample CRC-024T for the analysis shown in Figure 3 and state in the text (section 2.2) that they worked only with six CRC PDXs. Sample CRC-024T and its specific characteristics can be introduced in section 2.3 and Figures 4 and 5. Lines 150-166 need extensive revision for clarity. Response: We apologize for the confusion in data description and really appreciate the reviewer’s kind suggestion for the Section 2.2. and 2.3. including Figure 4 and Figure 5 for better understanding. Per suggestion, the data from only six CRC PDXs were demonstrated in Figure 3 (Section 2.2) and the findings associated with CRC-024T were demonstrated in the Section 2.3, Figures 4 and Figure 5.
â‘ Lines 147-185: To compare the effectiveness of GC1118 and cetuximab in vivo, six CRC PDXs (three KRAS wild-types and three KRAS mutants; all PIK3CA wild-type) were treated with GC1118 for at least 28 days (Figure 3). To evaluate the effects of GC1118 and cetuximab, tumor growth inhibition index (TGII) values were calculated from the average volume of the treated (Vt) and vehicle control (Vvc) groups using the following equation: TGII (%) = (Vt final -Vt initial)/(Vvc final -Vvc initial) × 100 [32]. For example, if the treatments resulted in no change in growth vs vehicle-treated controls, TGII (%) = 100. If GC1118 or cetuximab results in 70% tumor growth compared to vehicle-treated control tumors, TGII (%) = 300. Both GC1118 (TGII = −36.8%, P = 0.0053) and cetuximab (TGII = −29.4%, P = 0.006) induced complete tumor regression in CRC-003T PDXs (KRAS-wild-type; high-affinity ligand, 77.2%; low-affinity ligand, 22.8%) (GC1118 vs. cetuximab, P = 0.09; Figure 3A, upper panel and Supplemental Table 2). Similarly, treatment with GC1118 or cetuximab significantly inhibited CRC-077T growth (KRAS-wild-type; high-affinity ligand, 51.6%; low-affinity ligand, 48.4%) with TGII values of 6.8% (P = 0.0016) and 10.4% (P = 0.0015), respectively (Figure 3A, upper panel and Supplemental Table 2), suggesting comparable anti-tumor potency of GC1118 to cetuximab in patients harboring wild-type KRAS. Interestingly, GC1118 showed a significantly superior efficacy (TGII: 36.7%, P = 0.006) to cetuximab (TGII: 36.7%, P = 0.006) in CRC001T PDXs (KRAS-wild-type; high-affinity ligand, 88.7%; low-affinity ligand, 11.3%) (GC1118 vs. cetuximab, P < 0.001; Figure 3A, upper panel and Supplemental Table 2). Of note, GC1118 showed a more significant inhibitory effect on tumor growth than did cetuximab in cases of KRAS-mutant CRC (Figure 3A, lower panel and Supplemental Table 2). In CRC-026T PDXs (KRAS G12D; highaffinity ligand, 84.2%; low-affinity ligand, 15.8%), TGII values for GC1118 and cetuximab were 47.9% (P = 0.006) and 97.5% (P = 0.053)respectively (P = 0.001; Figure 3A, lower panel and Supplemental Table 2). Further, TGII values for GC1118 and cetuximab in CRC-034T (KRAS G12V; high-affinity ligand expression, 72.2%; low-affinity, 27.8%) were 34.5% (P = 0.023) and 103.6% (P = 0.12), respectively (GC1118 vs. cetuximab, P = 0.019; Figure 3A, lower panel and Supplemental Table 2). Finally, treating CRC-088T PDXs (KRAS G12V; high-affinity ligand, 61.3%; low-affinity ligand, 38.7%) with GC1118 and cetuximab resulted in TGII values of 10.8% (P = 0.001) and 47.6% (P = 0.91), respectively (GC1118 vs. cetuximab, P = 0.012; Figure 3A, lower panel and Supplemental Table 2). Overall, no significant differences were observed in the body weights of animals over the course of this study (Supplemental Figure 3).
â‘¡ Lines 196-229: The combined TGII from a panel of CRC PDXs revealed that GC1118 treatment inhibited tumor growth significantly better than cetuximab in KRAS-mutants (Figure 3 and Supplemental Table 2); however, complete tumor regression was not observed. In seven CRC PDXs with varying levels of basal EGFR, AKT, and ERK1/2 activation before GC1118 treatment (Figure 4A) including an additional CRC-024T model (KRAS G12D; high-affinity ligand, 88.8%; low-affinity ligand, 11.2%; high basal AKT activity) with resistance to GC1118 and cetuximab (TGII-GC1118 =65.6)(Supplemental Table 2), the efficacy of GC1118 (TGII) showed a significant positive correlation with basal AKT activity only (Pearson’s r = 0.82, P = 0.024) (Figure 4B). PI3K activity is the main predictor of MEK-inhibitor resistance in KRAS-driven CRC [33, 34] and thus, the additional use of a PI3K inhibitor could overcome resistance to MEK inhibition [35]. Although KRAS can directly activate PI3K signaling by binding to the p110-PI3K subunit, there is increasing evidence that PI3K activation following MEK inhibition is correlated with RTK activity, providing the foundation for the use of RTK inhibitors in KRAS-mutant CRC [36]. Based on these findings, we performed preliminary in vivo experiments evaluating the combination of GC1118 and the dual PI3K/mTOR inhibitor BEZ-235 [27] in a relatively GC1118-resistant CRC-024T model (KRASG12D showing high basal AKT activity (Figure 5). Here, cetuximab was inactive (TGII = 109.4%, P = 0.600), whereas GC1118 (TGII = 65.6%, P = 0.255) or BEZ-235 (TGII = 67.4%, P = 0.103) alone had moderate antitumor effects (Figure 5A and Supplemental Table 2). Further, the combination of the two molecules exerted significant inhibitory effects on tumor growth (TGII = 31.6%; P = 0.007; Figure 5A) with no reduction in body weight (Figure 5B) and without any other signs. We also confirmed significant inhibitory effects on AKT and ERK1/2 activity using IHC (Figure 5C and 5D) and immunoblotting (Figure 5E).
3. 4) From the data the authors present, I am still not convinced about a statistically significant predominance of GC1118 response over cetuximab, in KRAS W/T samples (Figure 3A, N=3, better GC1118 efficacy only in 1 out of 3). If the authors don’t have the means to provide additional data derived from a larger number of KRAS W/T samples, then it will better to focus their conclusions on the better efficacy of GC1118 over cetuximab ONLY in KRAS mutant PDXs. As such, any relevant statements should be avoided within the text and the title of 2.2 section should be altered accordingly. Response: We totally agree with the reviewer’s comments. Per suggestion, the title of Section 2.2 and some relevant statements in Abstract, Section 2.3 and Section 3 (Discussion) were changed. â‘ Lines 32-35: In a small panel of PDXs, GC1118 was more effective against in KRAS mutant tumors than cetuximab, whereas two agents demonstrated comparable efficacy against wild-type KRAS PDXs. Persistent phosphatidylinositol-3-kinase (PI3K)/AKT signaling was thought to underlie resistance to GC1118. â‘¡ Lines 193-194: 2.3. Activation of AKT signaling confers resistance to GC1118 monotherapy in KRAS-mutant CRC PDX models â‘¢ Lines 174-191: Of note, GC1118 showed a more significant inhibitory effect on tumor growth than did cetuximab in cases of KRAS-mutant CRC (Figure 3A, lower panel and Supplemental Table 2). In CRC-026T PDXs (KRAS G12D; high-affinity ligand, 84.2%; low
affinity ligand, 15.8%), TGII values for GC1118 and cetuximab were 47.9% (P = 0.006) and 97.5% (P = 0.053)respectively (P = 0.001; Figure 3A, lower panel and Supplemental Table 2). Further, TGII values for GC1118 and cetuximab in CRC-034T (KRAS G12V; highaffinity ligand expression, 72.2%; low-affinity, 27.8%) were 34.5% (P = 0.023) and 103.6% (P = 0.12), respectively (GC1118 vs. cetuximab, P = 0.019; Figure 3A, lower panel and Supplemental Table 2). Finally, treating CRC-088T PDXs (KRAS G12V; high-affinity ligand, 61.3%; low-affinity ligand, 38.7%) with GC1118 and cetuximab resulted in TGII values of 10.8% (P = 0.001) and 47.6% (P = 0.91), respectively (GC1118 vs. cetuximab, P = 0.012; Figure 3A, lower panel and Supplemental Table 2). Overall, no significant differences were observed in the body weights of animals over the course of this study (Supplemental Figure 3). The potent inhibitory effect of GC1118 on high-affinity EGFR ligand-induced signaling is more pronounced for downstream signaling molecules including AKT and ERK1/2 [8]. GC1118 and cetuximab resulted in variable inhibitory effects on ERK and AKT activation compared to that in the control group according to each PDX, as measured by IHC (Figure 3B, 3C and Supplemental Table 3) and immunoblotting (Supplemental Figure 4). Overall, ERK and AKT signaling activities were significantly suppressed after treatment with GC1118 alone compared to that with cetuximab alone. â‘£ Lines 267-275: Herein, we show that efficiently blocking high-affinity EGFR ligands with GC1118 induces superior therapeutic benefits in KRAS mutated CRC PDX platform refractory to cetuximab. In addition, basal up-regulated AKT pathway was correlated with lower efficacy of GC1118, and our preliminary, promising results indicated that GC1118 combined with the PI3K/mTOR/AKT inhibitor BEZ-235 showed improved antitumor effects on KRAS-mutant tumors with intrinsically high AKT activity with favorable safety, encouraging further studies using novel therapeutic combinations to treat clinically-aggressive KRAS-mutant CRC showing elevated ratios of high- to low-affinity EGFR ligands and PI3K/mTOR/AKT signaling (Figure 6). ⑤ Lines 363-372: In summary, the superior inhibitory activity of GC1118 on high-affinity EGFR ligands, for which current clinical antibodies show restricted inhibitory activity, reflects the potential therapeutic advantage of this drug for the treatment of cancer in which high-affinity EGFR ligands are implicated in tumor progression, metastasis, and resistance to current cancer therapeutics. Although future work should focus on the development of predictive biomarkers and hypothesis-driven rational combinations, GC1118 might be of therapeutic benefit, alone or in combination with other agents, for KRAS-mutant mCRCs with elevated ratios of high- to low-affinity EGFR ligands and intrinsic PI3K–AKT pathway activation. Further validation based on mouse trials is required based on an expanded CRC PDX panel to overcome the heterogeneity encountered in the clinic and optimize clinical trial designs and further define a patient enrichment strategy.
4. 5) I suggest any explanations and relevant references given about the lack of the proposed (by the reviewer) studies to be also included in the general discussion section. The authors
also mention in their response that ‘Unfortunately, the cases with the mutations in both KRAS and PIK3CA for in vivo validation was not available among our panel of CRC PDXs’’. However in the sample panel used for the study, there were 3 cases of both KRAS and PIK3CA mutants…
Lines 282-309: Constitutively active MAPK signaling in KRAS-mutated CRC promotes epithelial–mesenchymal transition and cancer stemness, independent of external EGFR stimulation [43, 44]. Further, persistent downstream signaling through the RAS axis due to KRAS mutations can activate multiple processes involved in tumor progression and metastasis without the influence of EGFR and other cell surface receptor kinases. Previous studies have reported a significant association between EREG/AREG expression and cetuximab response in KRAS-wild-type patients, but not in KRAS-mutant patients [6, 810, 20, 21, 45, 46]. Therefore, there is an unmet need for novel EGFR-targeting therapies as alternative treatment options. Our results show that CRC PDXs harboring KRAS mutations expressed remarkably higher levels of high-affinity EGFR ligands than KRASwild-type tumors, suggesting that the expression levels of EGFR ligands could be used as biomarkers to predict the therapeutic response to EGFR-targeting strategies. Although EREG and AREG are predominant EGFR ligands expressed in CRC, and only a small fraction of high-affinity ligands is expressed [29], upon downstream activation of the EGFR/RAS/MAPK axis owing to a mutated KRAS effector, the expression of AREG and EREG ligands would be biologically irrelevant in terms of any benefit from cetuximab [8, 20, 21, 45]. The observed superior anti-tumor potency of GC1118 over cetuximab against CRC PDXs harboring activating KRAS mutations could be due to the strong inhibitory activity of the interaction between EGFR and high-affinity EGFR ligands [8, 20, 21], providing a rationale for clinical application of the expression pattern of EGFR ligands as a novel biomarker predictive of the response to GC1118 in treating patients with refractory mCRC. Supporting our work, increased secretion of the high-affinity EGFR ligands TGF-α and BTC by some KRAS-mutant clones has been suggested to be a paracrine resistance mechanism to anti-EGFR antibodies in CRC models[47-49]. Considering the significant roles of high-affinity EGFR ligands in modulating the tumor microenvironment and inducing resistance to various cancer therapeutics, our study suggests potential therapeutic advantages for GC1118 in terms of efficacy and the range of patients for whom it will be beneficial. Genetic and molecular mechanisms determining
the ratio of high-affinity / low-affinity EGFR ligands other than KRAS mutation status should be elucidated through further comparative analyses of the therapeutic effects of GC1118 on CRC PDXs secreting mainly high- or low-affinity EGFR ligands using a larger panel of heterogenous CRC PDXs. â‘¡ Lines 310-332: Here, importantly, we found that resistance to GC1118 is associated with increased activation of AKT signaling, suggesting that persistent activation of the PI3K/AKT/mTOR signaling axis by high-affinity EGFR ligands could be a potential feedback and resistance mechanism inducing EGFR inhibition. Although we focused on CRC PDX cases harboring only KRAS mutations to validate the potential of combined PI3K/mTOR/AKT and EGFR inhibition in KRAS-mutant CRC cells with high AKT activity due to several mechanisms such as the ratio of high- to low-affinity EGFR ligands, further investigations on CRC PDXs harboring concurrent mutations in both KRAS and the genes activating PI3K/mTOR/AKT pathway (ex. PIK3CA) are required to strengthen the importance of PI3K/mTOR/AKT pathway in the resistance to GC-1118. Genetic mutations in the PI3K and MAPK pathways are frequently implicated in CRC [10-12]. CRC patients with PIK3CA and KRAS mutations are unlikely to respond to the inhibition of the MEK pathway alone or the PI3K pathway alone but will require effective inhibition of both MEK and PI3K/AKT signaling pathways [12, 13, 16, 34, 39, 50-55]. For example, BEZ-235 in combination with EGFR inhibitors is more effective for less mTOR inhibitor-sensitive and EGFR inhibitor-resistant CRC cell lines, especially HCT116 (which harbors KRAS and PIK3CA mutations), as shown in a recent study [39]. Previous findings suggest that acquired resistance to anti-EGFR MoAbs biochemically converges on RAS/RAF/MEK/ERK and PI3K/mTOR/AKT pathways, coupled with cross-talk mechanisms between other members of the EGFR family, such as HER2 and HER3, as well as IGF1R [39, 55-59]. Additionally, it is well established that autophagy is associated with resistance to anti-EGFR MoAb therapy because EGFR stimulates multiple downstream signaling pathways that affect autophagy, including the PI3K–AKT–mTOR axis [7, 60]. Combination therapy comprising anti-EGFR MoAbs together with autophagy-inducing PI3K/mTOR inhibitors could be used to develop an active therapeutic strategy for mCRC patients by inducing autophagic cell death [61, 62]. â‘¢ Lines 333-362: Activating mutations in PIK3CA (phosphatidylinositol-4,5-bisphosphate 3-kinase, catalytic subunit alpha) are present in 15–20% of CRCs, and the prevalence of PIK3CA exon 9 and/or exon 20 hotspot mutations increases continuously from rectal (10%) to cecal (25%) cancers, supporting the colorectal continuum paradigm [13, 14, 55, 63-73]. Coexisting PIK3CA and KRAS mutations, which occur in approximately 8–9% of CRC cases [55, 66-68, 73-78], predict resistance to anti-EGFR therapy, as well as worse prognosis, in CRC [16, 39, 52, 55, 66, 68, 76, 79-86]. Interestingly, mutations in PIK3CA exon 9 (and to a lesser extent exon 20) are associated with features of the traditional serrated pathway (CpG island methylator phenotype-low (CIMP-low)/KRAS mutation) of tumorigenesis [66, 68, 76, 78]. Insight into KRAS-driven CRCs will stimulate new research to find the best approach to treat this aggressive type of cancer, encouraging further evaluations of novel combination strategies including PI3K/mTOR/AKT inhibitors
[39, 56, 87]. Although only one case was tested in the present study, our data highlight the potential of combined PI3K/mTOR and EGFR inhibition for KRAS-mutant CRC cells with relatively high levels of high-affinity EGFR ligands, although further investigation on the therapeutic efficacy, mode of action, and tolerability of this combination based on additional KRAS-mutant PDX models concurrently harboring other genetic alterations (with different genetic backgrounds) is required. In fact, there were three cases with mutations in both KRAS and PIK3CA among our panel (CRC-017T: KRAS G13D, PIK3A Q546K, TP53 R81X and P27R; CRC-021T: KRAS G13D); however, they could not be used for in vivo validation due to the difficulty in obtaining enough number of PDX cells for in vivo combination efficacy test. The verification of synergy of GC1118 and BEZ-235 in several KRAS-mutant CRC PDX cases less susceptible to GC1118 by high AKT activity is very essential to provide clinical reliability and strong support for our hypothesis highlighting the potential of combined PI3K/mTOR and EGFR inhibition in KRAS-mutant CRC cells with relative high levels of high affinity EGFR ligands. Our data highlight the potential of combined PI3K/mTOR/AKT and EGFR inhibition in KRAS-mutant CRC cells with relatively high levels of high affinity EGFR ligands, with further investigation on the therapeutic efficacy, mode of action, and tolerability for optimizing this combination in additional KRAS-mutant PDX models concurrently harboring other genetic alterations being required. As the low frequency of these double-mutant cases underscores the need for collaborative international efforts to undertake such drug combination studies, optimizing the design of such clinical trials for CRC requires a detailed knowledge of the prevalence of these respective mutant genotypes.
5. 6) Similarly, the discussion about the raised question and its significance should also be included in the discussion section. Response: We thank the reviewer for making important suggestions. Per suggestion, during resubmission of second round revision, the comments including our response to previous major comment 6 were added in Section 1 (introduction) and Section 3 (discussion) for clear a clear understanding.
â‘ Lines 79-85: GC1118 is a human anti-EGFR IgG1 antibody that differs from existing antiEGFR MoAbs such as cetuximab and panitumumab in its constant region, affinity, mode of action, and efficacy [8, 20]. A recent first-in-human phase I study of GC1118 conducted on patients with refractory solid tumors including gastric cancer and CRC showed promising clinical antitumor efficacy and tolerability [22]. Notably, GC1118 exhibited superior inhibitory effects on high-affinity ligand-induced signaling in CRC and gastric cancer cells, regardless of KRAS status, triggering more potent antitumor activity than cetuximab and panitumumab [8, 20]. â‘¡ Lines 251-275: As CRCs differ in clinical presentation, molecular heterogeneity, and the involvement of several molecular pathways and molecular changes [5, 37], PDXs represent the fastest and most effective approach to uncover active therapeutic agents for CRC [24-26]. In contrast to previous studies, we utilized the PDX platform to evaluate the
efficacy of GC1118 and its mechanism of action, as the induction and expression of highaffinity EGFR ligands have been reported to be more prevalent in CRC tumor xenografts than in in vitro cultures [8]. GC1118 is a human anti-EGFR IgG1 antibody that differs from existing anti-EGFR MoAbs, such as cetuximab and panitumumab, in its constant region, affinity, mode of action, and efficacy [8, 20], exhibiting superior binding affinity (resulting in ADCC) to both the low- and high-affinity variants of FcgRIIIA compared to cetuximab [8, 20]. Moreover, the use of BALB/c nude mice with intact innate immune systems could allow for the evaluation of GC1118-mediated ADCC through Fc receptors present on immune effector cells such as macrophages, monocytes, and natural killer cells [8, 11, 38]. A subset of CRCs lacking KRAS pathway mutations and showing "EGFR addictio n" is treatable using two EGFR-targeting MoAbs, namely cetuximab and panitumu mab [4, 9]. When the oncogenic stimulus occurs downstream, such as in tumors with KRAS mutations, resistance to these therapies arises [4, 5, 7, 12, 16, 39, 40]. KRAS mutations in CRC are associated with a more rapid onset and aggressive metastasis, making it clinically more challenging [16, 41, 42]. Herein, we show tha t efficiently blocking high-affinity EGFR ligands with GC1118 induces superior the rapeutic benefits in KRAS mutated CRC PDX platform refractory to cetuximab. In addition, basal up-regulated AKT pathway was correlated with lower efficacy of GC1118, and our preliminary, promising results indicated that GC1118 combined with the PI3K/mTOR/AKT inhibitor BEZ-235 showed improved anti-tumor effects on KRAS-mutant tumors with intrinsically high AKT activity with favorable safety, encouraging further studies using novel therapeutic combinations to treat clinically -aggressive KRAS-mutant CRC showing elevated ratios of high- to low-affinity EG FR ligands and PI3K/mTOR/AKT signaling (Figure 6). â‘¢ Line 289-309: Our results show that CRC PDXs harboring KRAS mutations expressed remarkably higher levels of high-affinity EGFR ligands than KRAS-wild-type tumors, suggesting that the expression levels of EGFR ligands could be used as biomarkers to predict the therapeutic response to EGFR-targeting strategies. Although EREG and AREG are predominant EGFR ligands expressed in CRC, and only a small fraction of highaffinity ligands is expressed [29], upon downstream activation of the EGFR/RAS/MAPK axis owing to a mutated KRAS effector, the expression of AREG and EREG ligands would be biologically irrelevant in terms of any benefit from cetuximab [8, 20, 21, 45]. The observed superior anti-tumor potency of GC1118 over cetuximab against CRC PDXs harboring activating KRAS mutations could be due to the strong inhibitory activity of the interaction between EGFR and high-affinity EGFR ligands [8, 20, 21], providing a rationale for clinical application of the expression pattern of EGFR ligands as a novel biomarker predictive of the response to GC1118 in treating patients with refractory mCRC. Supporting our work, increased secretion of the high-affinity EGFR ligands TGF-α and BTC by some KRAS-mutant clones has been suggested to be a paracrine resistance mechanism to anti-EGFR antibodies in CRC models[47-49]. Considering the significant roles of high-affinity EGFR ligands in modulating the tumor microenvironment and inducing resistance to various cancer therapeutics, our study suggests potential
therapeutic advantages for GC1118 in terms of efficacy and the range of patients for whom it will be beneficial. Genetic and molecular mechanisms determining the ratio of highaffinity / low-affinity EGFR ligands other than KRAS mutation status should be elucidated through further comparative analyses of the therapeutic effects of GC1118 on CRC PDXs secreting mainly high- or low-affinity EGFR ligands using a larger panel of heterogenous CRC PDXs. â‘£ Lines 343-362: Although only one case was tested in the present study, our data highlight the potential of combined PI3K/mTOR and EGFR inhibition for KRAS-mutant CRC cells with relatively high levels of high-affinity EGFR ligands, although further investigation on the therapeutic efficacy, mode of action, and tolerability of this combination based on additional KRAS-mutant PDX models concurrently harboring other genetic alterations (with different genetic backgrounds) is required. In fact, there were three cases with mutations in both KRAS and PIK3CA among our panel (CRC-017T: KRAS G13D, PIK3A Q546K, TP53 R81X and P27R; CRC-021T: KRAS G13D); however, they could not be used for in vivo validation due to the difficulty in obtaining enough number of PDX cells for in vivo combination efficacy test. The verification of synergy of GC1118 and BEZ-235 in several KRAS-mutant CRC PDX cases less susceptible to GC1118 by high AKT activity is very essential to provide clinical reliability and strong support for our hypothesis highlighting the potential of combined PI3K/mTOR and EGFR inhibition in KRAS-mutant CRC cells with relative high levels of high affinity EGFR ligands. Our data highlight the potential of combined PI3K/mTOR/AKT and EGFR inhibition in KRAS-mutant CRC cells with relatively high levels of high affinity EGFR ligands, with further investigation on the therapeutic efficacy, mode of action, and tolerability for optimizing this combination in additional KRAS-mutant PDX models concurrently harboring other genetic alterations being required. As the low frequency of these double-mutant cases underscores the need for collaborative international efforts to undertake such drug combination studies, optimizing the design of such clinical trials for CRC requires a detailed knowledge of the prevalence of these respective mutant genotypes.
6. 7) Satisfactory revision, further improvement still needed. Figure 4A shows fold change of expression relative to sample CRC-001T. Why the CRC-001T is used as control? Figure 4A should present the actual expressions derived from WB relative to actin which serves as internal control. Response: Thank you for the comment. In Figure 4, basal activity levels of EGFR, AKT, and ERK1/2 pathways based on western blotting using tumor xenografts from each CRC PDX model. The tumor samples were isolated when the tumor xenografts reached 200 mm3. For quantification, images were acquired and signal intensity of each protein band was quantified using the ImageJ software (NIH, Bethesda, MD, USA) and normalized to ?-actin. The activities of EGFR, AKT, and ERK1/2 were determined by normalization with their total pairs, namely phospho-EGFR/EGFR, phospho-AKT/AKT, and phospho-ERK1/2/ERK1/2, respectively (b) Pearson's correlation analysis was performed to analyze the correlation between relative EGFR, AKT, and ERK1/2 activities (X-axis) and the tumor growth inhibition index (TGII, Yaxis)(refer Lines 208-214 (the legend for revised Figure 4A). Per suggestion, actual signaling
activities measured in each CRC PDXs instead of the ratio to CRC-001T were used in Figure 4A and 4B as well as Supplementary Figure 4.
7. 8) Given that the sample number is extremely small (N=1) and the authors don’t have the means to include additional samples in the study, any conclusion about a statistically significant synergistic effect GC1118 and NVP-BEZ235 on resistant CRC tumors, should be avoided. Instead they can refer to evidences derived from their preliminary findings. As such relevant statements within the text and the title of section 2.3 should be revised, accordingly. Response: Thank you for the comment. Per suggestion, relevant statements within the text and the title of Section 2.3 were revised.
â‘ Lines 193-194: 2 . 3 . Activation of AKT signaling confers resistance to GC1118 monotherapy in KRAS-mutant CRC PDX models â‘¡ Lines 216-229: PI3K activity is the main predictor of MEK-inhibitor resistance in K
RAS-driven CRC [33, 34] and thus, the additional use of a PI3K inhibitor could o vercome resistance to MEK inhibition [35]. Although KRAS can directly activate P I3K signaling by binding to the p110-PI3K subunit, there is increasing evidence th at PI3K activation following MEK inhibition is correlated with RTK activity, provi ding the foundation for the use of RTK inhibitors in KRAS-mutant CRC [36]. Bas ed on these findings, we performed preliminary in vivo experiments evaluating th e combination of GC1118 and the dual PI3K/mTOR inhibitor BEZ-235 [27] in a rel atively GC1118-resistant CRC-024T model (KRASG12D showing high basal AKT ac tivity (Figure 5). Here, cetuximab was inactive (TGII = 109.4%, P = 0.600), wherea s GC1118 (TGII = 65.6%, P = 0.255) or BEZ-235 (TGII = 67.4%, P = 0.103) alone h ad moderate antitumor effects (Figure 5A and Supplemental Table 2). Further, th e combination of the two molecules exerted significant inhibitory effects on tumor growth (TGII = 31.6%; P = 0.007; Figure 5A) with no reduction in body weight (F igure 5B) and without any other signs. We also confirmed significant inhibitory ef fects on AKT and ERK1/2 activity using IHC (Figure 5C and 5D) and immunoblo tting (Figure 5E). â‘¢ Lines 269-275: In addition, basal up-regulated AKT pathway was correlated with lo wer efficacy of GC1118, and our preliminary, promising results indicated that GC 1118 combined with the PI3K/mTOR/AKT inhibitor BEZ-235 showed improved ant i-tumor effects on KRAS-mutant tumors with intrinsically high AKT activity with favorable safety, encouraging further studies using novel therapeutic combinations to treat clinically-aggressive KRAS-mutant CRC showing elevated ratios of high- t o low-affinity EGFR ligands and PI3K/mTOR/AKT signaling (Figure 6). â‘£ Lines 333-362: Activating mutations in PIK3CA (phosphatidylinositol-4,5-bisphosphate 3-kinase, catalytic subunit alpha) are present in 15–20% of CRCs, and the prevalence of PIK3CA exon 9 and/or exon 20 hotspot mutations increases continuously from rectal (10%) to cecal (25%) cancers, supporting the colorectal continuum paradigm [13, 14, 55, 63-73]. Coexisting PIK3CA and KRAS mutations, which occur in approximately 8–9% of CRC cases [55, 66-68, 73-78], predict resistance to anti-EGFR therapy, as well as worse prognosis, in CRC [16, 39, 52, 55, 66, 68, 76, 79-86]. Interestingly, mutations in PIK3CA exon 9 (and to a lesser extent exon 20) are associated with features of the traditional serrated pathway (CpG island methylator phenotype-low (CIMP-low)/KRAS mutation) of tumorigenesis [66, 68, 76, 78]. Insight into KRAS-driven CRCs will stimulate new research to find the best approach to treat this aggressive type of cancer, encouraging further evaluations of novel combination strategies including PI3K/mTOR/AKT inhibitors [39, 56, 87]. Although only one case was tested in the present study, our data highlight the potential of combined PI3K/mTOR and EGFR inhibition for KRAS-mutant CRC cells with relatively high levels of high-affinity EGFR ligands, although further investigation on the therapeutic efficacy, mode of action, and tolerability of this combination based on additional KRAS-mutant PDX models concurrently harboring other genetic alterations (with different genetic backgrounds) is required. In fact, there were three cases with mutations in both KRAS and PIK3CA among our panel (CRC-017T: KRAS G13D, PIK3A
Q546K, TP53 R81X and P27R; CRC-021T: KRAS G13D); however, they could not be used for in vivo validation due to the difficulty in obtaining enough number of PDX cells for in vivo combination efficacy test. The verification of synergy of GC1118 and BEZ-235 in several KRAS-mutant CRC PDX cases less susceptible to GC1118 by high AKT activity is very essential to provide clinical reliability and strong support for our hypothesis highlighting the potential of combined PI3K/mTOR and EGFR inhibition in KRAS-mutant CRC cells with relative high levels of high affinity EGFR ligands. Our data highlight the potential of combined PI3K/mTOR/AKT and EGFR inhibition in KRAS-mutant CRC cells with relatively high levels of high affinity EGFR ligands, with further investigation on the therapeutic efficacy, mode of action, and tolerability for optimizing this combination in additional KRAS-mutant PDX models concurrently harboring other genetic alterations being required. As the low frequency of these double-mutant cases underscores the need for collaborative international efforts to undertake such drug combination studies, optimizing the design of such clinical trials for CRC requires a detailed knowledge of the prevalence of these respective mutant genotypes. ⑤ Lines 363-372: In summary, the superior inhibitory activity of GC1118 on high-affinity EGFR ligands, for which current clinical antibodies show restricted inhibitory activity, reflects the potential therapeutic advantage of this drug for the treatment of cancer in which high-affinity EGFR ligands are implicated in tumor progression, metastasis, and resistance to current cancer therapeutics. Although future work should focus on the development of predictive biomarkers and hypothesis-driven rational combinations, GC1118 might be of therapeutic benefit, alone or in combination with other agents, for KRAS-mutant mCRCs with elevated ratios of high- to low-affinity EGFR ligands and intrinsic PI3K–AKT pathway activation. Further validation based on mouse trials is required based on an expanded CRC PDX panel to overcome the heterogeneity encountered in the clinic and optimize clinical trial designs and further define a patient enrichment strategy.
Round 3
Reviewer 2 Report
The ms has been significantly improved after the second revision, therefore i suggest its publication in its present form